

# A mechanism of post-depositional processes affecting chlorine and its isotope in the upper snowpack of High Antarctic Plateau

Xavier Giraud[1], Mélanie Baroni[1], Rita Traversi[2]

[1]Aix Marseille Univ, CNRS, IRD, INRAE, Coll France, CEREGE, Aix-en-Provence, 13100, France

[2]Dept. of Chemistry Ugo Schiff, University of Florence, Florence, I-50019, Italy

*Correspondence to*: Xavier Giraud (giraud@cerege.fr)

**Abstract.** The main purpose of this work is to propose a mechanism of post-depositional processes affecting chlorine and its chlorine-36 cosmogenic nuclide in the upper snowpack of the High Antarctic Plateau. We suggest that the observed decrease of total chlorine content in the upper meters of the snowpack is due to a progressive release of the HCl content from ice. We

also propose a consistent framework, combining diffusion in bulk ice and snow microstructure. The observation of the low chlorine content in ice at depth leads to the robust hypothesis that the chemical equilibrium of chlorine between the ice and the snowpack interstitial air (SIA) is close to zero. HCl is thought to diffuse in ice, and to be progressively released in the SIA, and exported to the Antarctic atmosphere by the wind-ventilation. The time required to expel all the mobile species of chlorine (i.e., HCl) from snow depends on the diffusion coefficient of chlorine in ice combined with the snow grain size and its evolution

with depth. This work is synthesised in a model combining the microstructure evolution of the upper meters of a snowpack (changes in mean snow grain size) and the diffusion of chlorine in ice applied to single spherical grains. The variability observed in chloride concentration profiles with depth, at a same site but different sampling time or different snow pits, or among different sites of the High Antarctic Plateau, is mostly due to the variations in initial concentrations in HCl and sodium chloride (NaCl) species and the snow grain size evolution. This model offers a common framework for understanding the fate

of chlorine in Antarctica, from coastal to inland locations, including low accumulation sites on the plateau, far from the ocean. Applications of this post-depositional model to chlorine and to [36]Cl allows to picture a recycling mechanism of chlorine at the scale of Antarctica. In particular, the [36]Cl concentration in the surface snow of the Vostok site illustrates this recycling mechanism and the persistent contamination of inland Antarctica by anthropogenic [36]Cl originating from the marine nuclear tests of the 1950s to the 1970s.

**Introduction**

The surface snowpack of polar ice caps is the interface medium between the atmosphere and the firn. It is the location of chemical and physical processes that transform the snow record after it is deposited on the surface (Bartels-Rausch et al., 2014). These post-depositional transformations affect both the chemical signature that is buried over time and the atmospheric composition above the snowpack itself.



Concentrations of some compounds, among them volatile acidic species (chloride $Cl^-$, fluoride $F^-$, nitrate $NO_3^-$), decrease in the first meters and are thought to be expelled back to the atmosphere (Dibb and Jaffrezo, 1997; Wagnon et al., 1999). Post-depositional processes can occur in the very top surface (skin layer), where for example photochemical reactions affect bromine and contribute to Ozone Depleted Events (ODEs) (Toyota et al., 2014), and chemical species concentrations show a decrease as depth increases. Such a trend is well marked and relatively fast for nitrate, for instance, being probably a synergistic effect

of photolytic and condensation/evaporation processes (Bock et al., 2016; Akers et al., 2022), whereas it is more gradual for other components, like formaldehyde (HCHO) (Hutterli et al., 1999; Barret et al., 2011) or methanesulfonic acid (MSA) (Delmas et al., 2003), which are probably mostly affected by diffusion processes.

Understanding these post-depositional processes is important for interpreting ice core records for many purposes such as dating using the $^{36}Cl/^{10}Be$ ratio which is based on the principle of radioactive decay. Low accumulation sites, in Antarctica, are chosen

for their ancient ice. For instance, the site chosen in the frame of the European project Beyond-EPICA Oldest Ice is a few tens of kilometers away from the Dome C – Concordia station. If the $^{36}Cl$ content is altered, it might lower the $^{36}Cl/^{10}Be$ ratio and result in an erroneous age of the ice. The $^{36}Cl/^{10}Be$ ratio is also a proxy for evaluating the intensity of extreme solar particle events (e.g. Koldobskiy et al., 2022; Mekhaldi et al., 2015).

Snowpack models are usually physics-based and simulate the evolution of temperature, density of snow layers as a function

of weather conditions, including energy balance rules. Their degree of complexity varies according to their application, and they can include parameterizations for micro-structural characteristics (grain shape, connectivity…), metamorphism processes (compaction, sublimation, condensation…), and chemical species (halogens, isotopes, inert gases…).

Complex snowpack models tend to simulate realistic snowpack building on short time scales, compatible with operational forecasting (e.g. Brun et al., 1992). Models developed for alpine regions have been applied to polar environments: this is the

case for models such as SNOWPACK (Groot Zwaaftink et al., 2013) or CROCUS (Vionnet et al., 2012; Touzeau et al., 2018). Such applications are generally limited to time scales of no more than a few years and depend on realistic meteorological inputs.

Addressing chemical issues requires making ice-air interactions explicit. Ice composition may be considered either as a bulk

(Hutterli et al., 1999) or restricted to the surface of snow grains in a liquid-like layer (LLL) (Thomas et al., 2011). When chemical species are thought to migrate in solid ice, models include solid diffusion processes and must approximate the shape of the grain, usually as spherical grains (Barret et al., 2011; Bock et al., 2016; Touzeau et al., 2018). The evolution of nitrate (Bock et al., 2016; Chan et al., 2018) or HCHO (Barret et al., 2011) composition of the snowpack takes place in the upper layers (a few centimetres) or the so-called skin-layer (first few millimetres). Chemical processes can be very detailed, testing

parameterizations on adsorption and co-condensation of chemical species at the ice-air interface (Bock et al., 2016; Chan et al., 2018). Simulations typically cover short time scales, from days to years at most.





Chlorine content in the snowpack of the High Antarctic Plateau shows a similar decrease from the surface to depth but deserves specific considerations. The time scale of this concentration decrease spans over decades (see section 1.2 and Fig. 1),

suggesting slow processes. HCl is thought to diffuse into solid ice, but diffusion coefficients proposed by previous studies cover several orders of magnitude (see section 2.3.1). Considering that post-depositional and chemical reactions are acting permanently, we may also question whether the measured surface concentrations reflect the original content at the time of snow grain formations or are already transformed records.

The mobility of chlorine in the snowpack has also been questioned by (Delmas et al., 2004). They measured the cosmogenic nuclide of chorine-36 ($^{36}$Cl) in snow pits at Vostok, associated to the marine nuclear tests in the 1950s and 1960s. The authors observed an apparent shift of $^{36}$Cl profile compared to that of Cesium-137 ($^{137}$Cs), non-mobile in the snowpack and produced during atmospheric nuclear tests at the same time period. They suggest that the mobility of $^{36}$Cl (and chlorine) is the result of both diffusion mixing and advective transport. (Pivot et al., 2019) tested the advection-diffusion scheme proposed by (Delmas

et al., 2004), taking advantage of new high-resolution $^{36}$Cl data in a snow pit collected at Vostok in 2008, i.e. 9 years after the snow pit presented by Delmas and colleagues. Among the conclusions, it appears that the advection-diffusion scheme failed to explain the $^{36}$Cl signal shift while preserving the observed fine structures of the record. Furthermore, the mechanism behind a net advection term remains unexplained.

This work aims to propose a coherent physical mechanism explaining the evolution of chlorine in the upper meters of the Antarctic snowpack. We use the Vostok and Dome C sites as representative of the High Antarctic Plateau snowpack, characterized by low snow accumulation rates, among other things. These sites have been extensively studied and existing data provide important information on local variability of snow content. One challenge is to be able to combine the fine scale required to consider the solid diffusion into snow grains with the long timescale associated with stabilization of the record.

This apparent difficulty is an opportunity to constrain and propose values for the diffusion coefficient of HCl in snow grains.

## 1 Chlorine in Antarctica: from the state of the art to our model building

### 1.1 Origin of chlorine in Antarctica

In Antarctica, chlorine originates mainly from the ocean and the sea-salt release (brines)(Rankin et al., 2004). The chemical composition of surface snow at coastal sites (Dumont D'Urville, Talos Dome, ...) reflects the marine origin of chlorine, with a

chloride to sodium (Cl$^-$:Na$^+$) ratio close to 1.8, typical of sea-salt, essentially composed of sodium chloride (NaCl) (Legrand and Delmas, 1988; Iizuka et al., 2016). From the coast to the High Antarctic Plateau, the Cl$^-$:Na$^+$ ratio changes, both in atmospheric aerosols and in the surface snow, reflecting aging of aerosols due to chemical reactions of acidic species (nitric and sulfuric acids) on NaCl during the transport (Benassai et al., 2005). These chemical reactions lead to the release of gaseous HCl (Legrand and Delmas, 1988).





The chlorine concentration have been measured in the aerosols and in the gas phase at Concordia (Legrand et al., 2017). The Cl⁻:Na⁺ ratio in aerosols fluctuates between 0.5 and 1.9 according to the season, with a mean annual value of 0.7. The concentration of gaseous HCl fluctuates seasonally as well, suggesting either that HCl is more efficiently transported to Concordia than sea-salt aerosol during some time periods of the year or that it is re-emitted from the snowpack over the High Antarctic Plateau.


The surface snow composition at inland sites should be the result of dry depositions (aerosols with sea-salt chlorine with low Cl⁻:Na⁺ ratios) and wet depositions condensing gaseous species like HCl. In addition, the snow composition should be affected by inputs from drifting snow, formed elsewhere, and by post-depositional chemical processes in the snowpack.

**1.2 Post-depositional release of chlorine from the snowpack**

The chemical composition of surface snow at inland sites (Vostok, Dome C, …) shows that the total chloride concentration ranges between 100 and 180 ng.g⁻¹ and the sodium concentration between 20 and 40 ng.g⁻¹ (Fig. 1) leading to a Cl⁻:Na⁺ ratio above 4, and even 25 in certain snow layers, values that are much higher than the typical marine ratio of 1.8 or the ratio into dry aerosols (~0.7) (Legrand et al., 2017). It has been evidenced that the chlorine content in the snowpack at low accumulation sites, in the High Antarctic Plateau, is affected by post-depositional processes (Legrand and Delmas, 1988), leading to re-
emission of chlorine into the atmosphere and contributing as an additional source of gaseous HCl (Legrand et al., 2017) superimposed to that formed from chemical reactions on sea-salt aerosols.

The evolution of the total chlorine content of snow in the upper meters of the snowpack shows the same pattern at Vostok and Dome C sites (Fig. 1). The total chlorine concentration is high at the surface and decreases with depth. Upon reaching a certain depth (~3 m depth in the case of Vostok, Pivot et al., 2019), the Cl⁻:Na⁺ ratio is as low as 0.7 meaning the total chlorine content has decreased from 180 ng.g⁻¹ at the surface to 20 ng.g⁻¹ at 3 m depth.

Knowing the origin of chlorine is useful to distinguish the chemical species of chlorine. In a simplified view, the total chlorine content in the snowpack has two main sources: (i) the sea-salt contribution identified by the sodium (Na⁺) content: the sea-salt-related chlorine content (ssCl) is estimated as the Na⁺ concentration multiplied by the average Cl⁻:Na⁺ ratio in dry deposits,
i.e. 0.7 in our case study (ii) gaseous chloride assumed to be HCl. In regard to the Cl⁻:Na⁺ ratio of sea-salts, any value of this ratio higher than 1.8 would indicate an enrichment of HCl. In the case of the inland locations in the High Antarctic Plateau, considering the Cl⁻:Na⁺ ratio in dry aerosols, we will assume that any value of this ratio higher than 0.7 indicates the presence of HCl. Assuming that ssCl is non-mobile and non-reactive, the decrease of the chloride concentration in the snowpack can be attributed to a loss of the HCl component, which underwent post-depositional processes and a progressive release from the
snowpack. Given the low snow accumulation rates (~6 cm.yr⁻¹ at Vostok (Pivot et al., 2019), ~8 cm.yr⁻¹ at Dome C (Picard et al., 2019)), this release process takes a few decades to deplete most of the chlorine content of the snow. This approach is a




first-order description intended for the model and it would be refined by considering the full ionic composition of the snow and seasonal fluctuations in aerosol and gas phase compositions.

The chlorine budget is different at coastal sites, where the Cl⁻:Na⁺ ratio is constant from the surface to the depths covering actual climatic conditions and is close to 1.8 (Pivot et al., 2019; Benassai et al., 2005). Therefore, chlorine is mainly associated with sea-salt compounds with a low or no HCl contribution.

### 1.3 Ice-air exchange with the snowpack interstitial air

Considering the export of chlorine out of the snowpack requires identifying the pathways of this export. The snowpack is a
medium that can be thought as consisting of agglomerated snow grains and interstitial air (e.g. Calonne et al., 2017). The snowpack interstitial air (SIA) is assumed to be well ventilated until ~13 m depth at Vostok (Severinghaus et al., 2010), through wind pumping (Colbeck, 1989). Our study focuses on processes occurring in the top 3-5 meters of the snowpack. Therefore, in the following, we will consider the SIA content to be homogeneous over that depth interval, in connection with the free atmospheric air and reflecting the atmospheric composition. In this condition, it can be inferred that only the snow reservoir
changes its composition with depth. From a chemical point of view, we can ask whether the surface snow or the deep snow (3 m deep) are in equilibrium with the air composition of the SIA. If both the surface and deep layers are in equilibrium with the atmosphere, then the conditions of that equilibrium may have changed with depth. This would be possible by invoking changes in chemical conditions such as snow acidity. If the snow-air exchange conditions are similar between surface and deep layers, it is questionable whether any of them are in equilibrium. Deep snow has experienced snow-air exchange for decades and is
more likely to reflect this equilibrium. The total chlorine content in the deeper layers has very low values, such that the Cl⁻:Na⁺ ratio is less than one (Pivot et al., 2019).

One option for reconciling surface and depth conditions is to recall that snow grains are rounded ice grains (in the following, the use of the word "ice" is referring to the solid crystal phase and not to the deep transformation of snow after compaction
and metamorphism), and that total chlorine content is measured on bulk snow, i.e., bulk ice. We will consider that the distribution of HCl in the snow grain can be heterogeneous between the surface and the interior of the grain. The snow-air equilibrium must be considered as ice-air equilibrium at the surface of the snow grains.

Since the snow at depth has lost most of its chlorine, it implies that the ice-air equilibrium value for the mobile and volatile
form of chlorine (HCl) at the surface of the snow grain is close to zero. This could be explained by a weak affinity of HCl for solid ice in the presence of other acidic species, such as sulphuric acid (Hynes et al., 2002; Abbatt, 2003; Cox et al., 2005). Furthermore, experiments for determining the diffusion coefficient in ice (Dominé et al., 1994; Thibert and Dominé, 1997) indicate that HCl can migrate and be located in ice. HCl is therefore not restricted to the surface of snow grains, as it is proposed for other chemical species (nitrogen oxides, reactive bromine) to be localized in a liquid-like layer (LLL) (Thomas et al., 2011).





We hypothesize that the snow grain surface may be in chemical equilibrium with the SIA, i.e., close to zero HCl concentration, while the interior contents retain higher chlorine concentrations. The connection between the interior and surface of snow grains involves diffusion into a crystalline matrix (bulk diffusion) or along grain boundaries (Huthwelker et al., 2006). The mass transfer of chlorine from ice to air is thus the combination of diffusion into solid ice and ice-air exchange. Since we assume a well-ventilated SIA, the balance between the diffusion coefficient and the ice-air exchange rate will determine which

process is limiting. The time for chlorine to be released will be modulated by the combination of the diffusion coefficient and the mean size of the snow grains.

**1.4 Snow grain size at Vostok and Dome C**

The ice lattice of the snowpack layers consists of snow grains of different shapes, bound together with some degree of compaction (Hagenmuller et al., 2014). Snow grain shapes are described in many different contexts, from mountain glaciers

to taiga steppes or polar snowpacks (Colbeck, 1987; Legagneux et al., 2002). In the case of High Antarctic Plateau sites, such as Vostok or Dome C, the snowpack consists of rounded snow grains, transported and eroded by the wind, and surface hoar crystals (Gallet et al., 2011).

Among the parameters to quantify the microstructure of snow layers, the average size of snow grains is estimated by measuring

the specific surface area (SSA). SSA is the ratio of the surface area of the ice-air interface to the ice mass, expressed in square meters per kg. If the snowpack is considered to consist of individual uniform spherical snow grains of radius $R_{eff}$, SSA and $R_{eff}$ can be related by the ice density, $\rho_{ice}$, according to the formula in (Gallet et al., 2011):

$$R_{eff} = \frac{3}{\rho_{ice}\,\text{SSA}}\,. \tag{1}$$

Measurements of the SSA of precipitation particles in summer at Dome C range from 90 to 120 $m^2.kg^{-1}$ (Libois et al., 2015). SSA in the upper 2 mm of the snowpack generally decreases within a few days from 80 to about 30 $m^2.kg^{-1}$ under the effect of wind drift or meteorological conditions. Gallet et al. (2011) report SSA of about 38 $m^2.kg^{-1}$ in the first centimeter at Concordia station. The mean radius of snow grains is then around 0.1 mm following equation (1) and assuming a density of ice of 917 $kg.m^{-3}$.


SSA and snow density show a large variability from layer to layer, and follow a general evolution with depth with decreasing SSA (increasing mean grain size) and increasing density (Gallet et al., 2011; Hörhold et al., 2011; Proksch et al., 2015; Calonne et al., 2017; Weinhart et al., 2020), under the influence of gravity compaction, wind compaction, metamorphism. Note that if these processes operate without changing the total mass of the snow layer, the associated microstructure changes should not

affect the average bulk-ice concentration.



As a result, the mean size of snow grains increases with depth. At 70 cm depth at Concordia station, the SSA is less than 14 m².kg⁻¹, which is equivalent to a mean grain size of 0.23 mm (Gallet et al., 2011). Data from the ITASE expedition report a mean grain size of at least 0.6 mm at 7 m depth, and 0.8 mm and greater at 20 m depth (Linow et al., 2012). Figure 2 reports
the surface values of grain size calculated from SSA measurements as reported by Gallet et al. (2011) at Dome C, and the mean snow grain radius as a function of depth used in this work (see section of model description).

In a single snow layer, grain sizes cover a wide range, and the shape of snow grains is not perfect spheres. Furthermore, the evolution of the mean snow grain size is not monotonic with depth, and high-resolution measurements show large variability in both density and SSA from layer to layer (Hörhold et al., 2011; Proksch et al., 2015). Therefore, the formula of the grain
radius proposed in our model and shown in Figure 2 is a convenient average of its evolution with depth and is not the result of a fine description of the processes responsible for this evolution.

## 2 Model description

In our model, we have considered the fate of chlorine in the surface snowpack of the High Antarctic Plateau as follows: snow grains are deposited on the surface of the snowpack with high chloride concentrations of about 150 ng.g⁻¹ or more, most of
which is in the form of HCl, and solid sea-salt compounds with a Cl⁻:Na⁺ ratio close to 0.7, corresponding to the ratio in aerosols. This snow composition, including HCl and other acidic species, is not in equilibrium with the interstitial air, and leads to the export of HCl, from the surface of snow grains to the SIA. The imbalance in concentration within the grain is compensated by bulk diffusion. Compaction and aggregation of snow grains accompanying burial results in an increase of mean grain radius with depth, which may delay the release of HCl to the SIA. Finally, the wind-ventilation of the SIA ensures
the re-emission of HCl from the snowpack to the atmosphere. It is also possible that the chlorine in the sea-salt still undergoes a chemical reaction in the snowpack and contributes to the release of HCl to the atmosphere (Legrand et al., 2017). This process is not considered in the present model.

### 2.1 General setting of the model

The model consists in following the evolution of the HCl content of individual snow grains during the burial process in the
snowpack. We consider that each layer consists of similar spherical snow grains, whose radius is the average radius calculated from the SSA and equation (1). Therefore, simulating the concentration in a single snow grain is representative of the entire corresponding snow layer.

Simulating the post-depositional mechanical processes responsible for the transformation of the microstructure is beyond the scope of this study. Therefore, the mean grain radius, $r$, as a function of depth, $z$, is prescribed according to the following
empirical equation:



$$r(z) = a\,z + b + \frac{(r0 - b)\,d}{z + d} \tag{2}$$

where $r0$ is the initial grain size at the time of deposition. The parameters a, b and c are empirical values to accommodate both surface and in depth grain sizes, as reported by (Gallet et al., 2011) and (Linow et al., 2012), respectively (Fig. 2). The formula proposed by (Linow et al., 2012) is not convenient for representing the surface values as reported by (Gallet et al., 2011), and

our empirical equation combining a linear trend and an asymptotic increase is convenient for testing the effect of different initial grain sizes or growth rates.

Each layer is also characterized by its thickness and deposition time interval. At time of deposition, the ratio of thickness to time interval is the accumulation rate. The chlorine content of sea-salt in the snow can also be prescribed for each layer, to track the total $Cl^-:Na^+$ ratio with time, but no chemical reaction affecting sea-salt chlorine is included in our model.

**2.2 Diffusion and ice-air exchange of chlorine**

For simplicity, we assume that spherical snow grains are formed with a homogeneous distribution of HCl, with an initial concentration $C_0$. In such an ideal case, analytical solutions for the diffusion in a sphere are proposed by (Crank, 1979), predicting the evolution of the total concentration with time, and considering different boundary conditions at the surface of the sphere. The choice of the boundary conditions at the surface of the sphere corresponds to approximations of the real

processes of chemical species exchange at the ice-air interface. A review of current knowledge can be found in (Huthwelker et al., 2006) and the possible processes cover surface accommodation, adsorption, dissociation, diffusion at the surface, and include the possible presence of a liquid like layer (LLL).

If the rate of ice-air exchange is rapid relative to other transfer processes, including the rate of internal diffusion or the

advection-diffusion in the SIA, then the concentration at the surface of the sphere can be considered in equilibrium with the air composition and constant. In this case, the practical boundary condition is to set a constant surface concentration, and the mean concentration in the sphere, C(t), as a function of time, t, is given by the following equation:

$$\frac{C(t) - C_0}{C_\infty - C_0} = 1 - \frac{6}{\pi^2} \sum_{n=1}^{\infty} \frac{1}{n^2} \exp\left(-D n^2 \pi^2\, t/r^2\right) \tag{3}$$

where $C_0$ is the initial mean concentration, $C_\infty$ is the final equilibrium concentration and D is the diffusion coefficient. In the

case where the kinetics of the ice-air exchange is sufficiently slow compared to the general mass transfer, it corresponds to the surface evaporation condition, that requires fixing the exchange rate. Analytical solutions can also be found in (Crank, 1979). The following results assume a constant surface concentration, but the evaporation condition is discussed in section 3.2 and recall in Appendix A.

Equation (3) is made for a sphere with a constant radius. In our model, the radius of the grains varies with time and depth (section 2.1 and Fig. 2), and we consider the grains growth as a reorganization of the ice mass, without modification of the





mean chlorine concentration nor ice mass addition. Therefore, when the grain size increases, it is possible to use equation (3) to compute the evolution of the mean concentration, but we must reevaluate the equivalent initial concentration to avoid adding artificial mass content. The principle is illustrated in Figure 3 for a simplified theoretical case. Starting with a snow grain of

0.2 mm radius and an initial concentration of 230 ng.g$^{-1}$, the mean concentration decreases during the burial (turquoise line). Let's increase the mean radius of the grains from 0.2 to 0.4 mm when it is buried at 0.5 m depth. At this point, the mean concentration of the grain is ~100 ng.g$^{-1}$. The future evolution of the snow grain can be inferred using equation (3) and the new radius (0.4 mm), but it requires to compute the virtual initial concentration: this can be done since all other parameters in equation (3) are known, and the virtual $C_0$ is ~150 ng.g$^{-1}$ (yellow line). The resulting trajectory passing from a 0.2 to 0.4 mm

radius is the green line in Figure 3. Of course, this approach is not an exact solution since the distribution of the chemical species in the grain would be affected by the reorganization of the grain shape and probably does not fit to the theoretical distribution. Nevertheless, the change in radius of the grain is gradual and provides a smooth continuum between the exact solutions, as shown in the result section about the effect of a changing radius.

## 2.3 Parameters

### 2.3.1 Diffusion coefficient

The diffusion coefficient of HCl in ice is a key parameter of this model, but former studies propose a large range of possible values. The review by (Huthwelker et al., 2006) reports diffusion constants covering many orders of magnitude, from $10^{-19}$ to $10^{-12}$ m$^2$.s$^{-1}$. Values above $10^{-11}$ m$^2$.s$^{-1}$ are reported but are likely to be affected by other conditions like ice melting. As shown in the result section, the time for chlorine to be released from the snow layers (a few decades) in combination with the size of

the snow grains (from ~0.1 to ~1 mm) will be a strong constraint for the choice of the diffusion coefficient. The optimal value used in our model is in the lower range of the reported ones, with D ~ $10^{-17}$ m$^2$.s$^{-1}$.

### 2.3.2 Initial concentration and age of the snow grains

The initial homogeneous concentration of snow grains, $C_0$, must be considered at the time of snow grain formation. It must also be considered that snow grains that are deposited at one site may have been formed abroad and transported by various

processes such as wind drift before being finally settled (e.g. (Picard et al., 2019)). This means that the snow grains are already aged and may have been affected by the release of part of their chemical content depending on the atmospheric conditions encountered. A simple option is to consider that the diffusion in the grain and the ice-air exchange already modify the concentrations of the grain during transport, under similar rules as those occurring in the snowpack, i.e., with the same equilibrium concentration $C_\infty$. During this pre-deposition phase, the size of the snow grains is held constant.

Therefore, at the time of deposition, the mean snow grain concentration, $C_{dep}$, may be different from $C_0$. The effect of considering a time lag, $t_{lag}$, between the formation of snow grains and their deposition is described in the section 3. In this case,



$C_{dep}$ should correspond to the observed concentration in the surface snow layers and $C_0$ is a theoretical initial concentration that should be related to chemical conditions where and when the snow grain was forming.

## 3 Results and sensitivity test at Dome C

### 3.1 Parameters optimization and results

In this section, we explore how the different parameters of the model influence the evolution of the mean concentration in the snow grain during its burial. The following simulations cover a depth of 6.5 m assuming an accumulation rate of 8 cm.yr$^{-1}$, close to observations at Dome C. Model results are compared to the set of data presented by (Traversi et al., 2009). This dataset has the advantage of highlighting the variability of total chlorine concentration in snow layers between snow pits sampled at

the same site (Dome C) but over several years (4 campaigns between 1997/98 and 2005/06, Fig. 1). The HCl content of each profile is estimated assuming a Cl$^-$:Na$^+$ ratio of 0.7 for ssCl, so that HCl = Cl$^-$ - 0.7 * Na$^+$.

In the default configuration, the ice-air equilibrium concentration $C_\infty$ is set to zero, there is no time lag before deposition, and the grain radius increases with depth following equation (2) (Fig. 2). The optimized parameters for D and $C_0$ are obtained by

minimizing a cost function based on the RMSD (root mean square of differences) based on HCl concentrations between the model and the data (See Appendix B and Fig. S1).

The model-data agreement is best for a diffusion coefficient D set to $10^{-17}$ m$^2$.s$^{-1}$, and the initial concentration $C_0$ is set to 215 ng.g$^{-1}$. The resulting concentrations in snow layers from the surface to 6.5 m depth is shown on Figure 4 (green line,

standard simulation) and can be compared to the data concentration profiles from (Traversi et al., 2009). The first result is that the optimum diffusion coefficient that we determine is compatible with the range of reported values (from $10^{-19}$ to $10^{-12}$ m$^2$.s$^{-1}$, see (Huthwelker et al., 2006) and reference in there). Unless otherwise noticed, these are the default parameters used in the following sensitivity tests (see Table 1 for a summary of the simulations setup).

### 3.2 Sensitivity tests

Figure 4 shows the effect of considering different evolutions of the grain radius with depth. For a constant grain radius, the evolution of the mean concentration corresponds to the analytical solution provided by (Crank, 1979) and tends toward zero (the value of $C_\infty$). (Fig. 4; brown lines: constant radius of 0.1 mm; thin black line: exact analytical solution). As expected, the decrease of the mean concentration with time is slower for larger snow grains (Fig. 4; brown lines: constant radius of 0.1 mm; turquoise lines: constant radius of 0.4 mm). For a spherical snow grain with a constant radius of 0.4 mm, it takes about 22

years to release 50% of the initial content, and less than 2 years if the radius is 0.1 mm. As the radius of the snow grain increases with depth, the rate of HCl release slows down, so that the time of total release is delayed compared to the case of constant radius. The standard simulation corresponds to the grain radius evolution presented in Fig. 2, and results in a concentration



profile that fits within the variability of the data records (Fig. 4; green lines: standard simulation; thin blues lines: HCl records of the 4 snow pits from (Traversi et al., 2009)). Lower values of the data records could correspond to snow layers that have

retained smaller grain sizes (Fig. 4; orange lines: moderate grain size increase). On the other hand, depths showing higher concentrations of HCl could correspond to layers where the grain size is larger. This is illustrated by two simulations: either the grain size increases progressively (Fig. 4; red lines: amplified grain size increase) or it could experience a drastic compaction in the early stages (Fig. 4; thick black lines: the mean grain size is already at 0.4 mm at the surface).

The effective diffusion coefficient in the ice lattice is determined by the intrinsic diffusion coefficient in the ice crystals, but also by the local arrangement of polycrystalline structures which could orient, accelerate, or slow down the mobility of molecules, the existence of inter-grain diffusion pathways, the inclusion of impurities. It is therefore possible that the effective diffusion coefficient varies from one snow grain to another. Figure 5 shows the effect of considering different values for the diffusion coefficient, by testing 3 values spanning almost one order of magnitude: $5 \ 10^{-18}$, $10^{-17}$ (the standard case) and $2 \ 10^{-17}$

$m^2.s^{-1}$. The effect of these different values on the rate of change of chlorine content is as important as considering the variability of the mean grain radius (see Fig. 4) and makes it a non-negligible parameter.

Figure 6 shows the effect of the initial concentration of chlorine in snow grains, $C_0$, by considering 3 values: 100, 215 (the standard case) and 300 $ng.g^{-1}$. This initial concentration may reflect the atmospheric conditions at the time and place of

formation of the snow grains, including the gaseous HCl concentration in the atmosphere as well as other physical and chemical conditions (temperature, presence of other acidic species…). The large amplitude tested for the surface concentrations (from 100 to 300 $ng.g^{-1}$) is rapidly reduced to a difference of ~45 $ng.g^{-1}$ at one meter depth between the two simulations. It appears that the variability in the initial concentration must be accounted for but is quickly damped and minored in comparison to other sources of recorded fluctuations.


Finally, among the 4 important parameters driving the fate of chlorine content, the last one is the time lag between the formation of the snow grains and the definitive deposition at the surface of the snowpack, tlag, given in months. We assume that during this period the grain radius is kept constant and that the HCl can diffuse and be released with the same conditions as in the snowpack. The concentration at deposition time, $C_{dep}$, is therefore different from the initial concentration.


Figure 7 illustrates that it is possible to find different combinations of initial concentration, $C_0$, and time lag before deposition, tlag, to achieve similar concentrations at deposition time, $C_{dep}$, and similar evolutions of chlorine content with depth. Note that for the simulations with a time lag of 4, 8 and 12 months, the optimum diffusion coefficient found is at $1.1 \ 10^{-17} \ m^2.s^{-1}$, $1.15 \ 10^{-17} \ m^2.s^{-1}$, $1.2 \ 10^{-17} \ m^2.s^{-1}$, respectively, slightly higher than for the default case (see Table 1). The major difference between

the four simulations with different time lag (presented in Figure 7) lies in the distribution of the chlorine content into the grain.





This is illustrated in Figure 8, that shows that for similar mean chlorine concentrations, the distribution may be more depleted on the outer sphere for simulations with higher time lag.

## 4 Application to the chlorine isotopes at Vostok

The model is applied to the site of Vostok, in order to interpret the chlorine records from (Pivot et al., 2019). The basic assumption is that fluctuations of the chlorine record around a mean decrease can be explained by differential grain size evolutions from layer to layer. A second hypothesis is also tested, considering the variations of initial snow concentration at deposition.

The accumulation rate is set to 6 cm.yr$^{-1}$, and we apply the coefficient diffusion of chlorine in ice found in the previous section, i.e. $10^{-17}$ m$^2$.s$^{-1}$. Considering that the site of Vostok may be similar to Dome C, a first guess simulation applying the standard grain size evolution used previously is run, and we find that the initial concentration of snow deposits would be ~440 ng.g$^{-1}$. (Fig. 9; green lines). The model simulates the fate of HCl only. The model-data comparison therefore focuses only on the HCl component of the data record, which is estimated by subtracting the sea-salt component (NaCl * 0.7) from the total chlorine.

The difference between the model result (Fig. 9c; green line) and the data (Fig. 9c; blue line) is used to reevaluate the mean grain size history of each layer. In Equation (2) setting the grain size evolution with depth, the coefficient b is mostly driving the growing rate of grains, as already seen about the sensitivity test of section 3.1 and Figure 4 (see also Table 1). If the data value is higher or lower than the model result, the coefficient b is proportionally increased or decreased, respectively. Similarly, and separately, the difference between model results from the first guess and the data are used to reevaluate the initial snow concentration $C_0$. Both simulations to reconstruct the grain size or $C_0$ are shown in Fig. 9., and both succeed to simulate the chlorine content in each layer, fitting the data (Fig. 9c).

When reevaluating the mean grain size history, the grain size of the different layers generally increases with depth and shows variations from layer to layer, with a maximum of ~0.6 mm (Fig. 9b). This variability in snow grain size history across layers is perfectly consistent with the large variability shown by other parameters observed in the upper snowpack, such as density (Hörhold et al., 2011). When reevaluating $C_0$, values fluctuate between ~250 and ~870 ng.g$^{-1}$ (Fig. 9a). This variability is more difficult to support but it cannot be excluded since we know too little about the chemical conditions of the snow grain formation.

Below 2.8 m depth, the snow layers are almost completely depleted in HCl. Somehow, the initial surface signal is lost, and its reconstruction becomes difficult. Inferring the initial snow grain concentration suggests HCl-free deposits as well (Fig. 9a).



This is in strong contrast with the reconstructed initial concentration levels for the rest of the record. It is difficult to envision that the periods of deposit corresponding to these deep layers have experienced chlorine-free snow deposits.

Alternatively, inference of the grain size history suggests that these deeper layers retained small mean grain sizes, compared to upper layers with larger snow grains. The model can easily explore the option of a smooth progression of the grain size history for each layer, but we could obtain equivalent results by considering a non-uniform progression. For example, the deep depleted layers might have kept small grains for a few years after deposition, promoting full release of HCl, before undergoing compaction like all other snow layers and resulting in larger mean snow grains. Such a scenario makes it possible to combine
both low-chlorine contents with large grain sizes.

To highlight the limits of these reconstructions, the high concentrations observed in some layers could result from the combination of a relatively low initial concentration, with an early snow aggregation event (deep icing, melting, condensation of water vapor) that would drastically slow the release of chlorine compared to other layers.
One of the highlights of (Pivot et al., 2019) was to present high-resolution data for chlorine and its cosmogenic nuclide $^{36}$Cl, at Vostok site over the period of the anthropogenic nuclear production. Here we apply the present model to the $^{36}$Cl nuclide, assuming that this deposit was only as H$^{36}$Cl (no Na$^{36}$Cl sea-salt anthropogenic contribution). We take advantage of the previous simulation of the total Cl (as the stable isotopes of $^{35}$Cl and $^{37}$Cl), which uses a constant initial HCl concentration $C_0$,
and provides the mean grain size in each layer. Comparing the resulting concentration at each depth with the initial concentration, it allows to compute the decrease proportion between the HCl deposit for each layer. Assuming that H$^{36}$Cl follows the same processes and knowing the final $^{36}$Cl concentration in each layer, it is therefore possible to reconstruct the H$^{36}$Cl concentration in snow deposit (see Appendix C for formulas). The Figure 10 shows the results, over two y-scales to be able to distinguish the peak values between 1957 and 1967 on one hand (Fig. 10a) and the long-term decrease on the other
hand (Fig. 10b).

The general pattern of the H$^{36}$Cl concentration in the snow deposit is very similar to the $^{36}$Cl:Cl ratio in the record presented by (Pivot et al., 2019) (Fig. 10c). This is expected since all nuclides (cosmogenic and stable) H$^{36}$Cl, H$^{35}$Cl and H$^{37}$Cl undergo the same release processes, and no other loss terms are involved (no advection or diffusion). The content of the snow deposit
shows a decrease from ~1968 to 2008 that is remarkably linear (Fig. 10a), with some variability, which is also reflected in the $^{36}$Cl:Cl ratio of snow layers (Fig. 10c). It follows that the final $^{36}$Cl content in the snow layers can be easily explained by combining the chlorine input, the $^{36}$Cl to Cl ratio of that input, and the modulation of the content (of both $^{36}$Cl and Cl similarly) by the grain size evolution.

The presence of two extremes around 1962-1969 and 1979 in the $^{36}$Cl record could have been interpreted in terms of the
advection-diffusion mechanism proposed by (Delmas et al., 2004). Instead, our simulations indicate that these fluctuations





could either be explained by higher values of the initial total chlorine concentration (Fig. 9a; orange curve), or more likely by larger grain sizes into these layers (Fig 9b, purple curve). In both cases, the linear decrease of the $^{36}$Cl to Cl ratio in the deposit is kept undisturbed, and most importantly no mobility of the chlorine and $^{36}$Cl among layers is invoked.

Variability in the snow content of chlorine at deposition probably exists and explains part of the record variability. However, if this would be the only source of variability, then it would require unrealistically high concentrations (Fig. 9). According to the proposed mechanism, the influence of the grain size variability is the main driver for the content variability among snow layers.

An unexpected result of this reconstruction is to distinguish a net peak value in the 1952 layer, which is the first year for major nuclear tests leading to the release of $^{36}$Cl in the stratosphere (Pivot et al., 2019). This peak was undistinguishable in the raw data of the record ($^{36}$Cl, total Cl$^-$, Na$^+$; Fig 10c and 10d), but could be found in the deposit reconstruction (Fig. 10a and 10b) by assuming that the deposition of $^{36}$Cl is done as H$^{36}$Cl (see Appendix C for formulas). We think that the identification of this peculiar year supports this latter assumption.

**5 Discussion**

The duration required for releasing the HCl from ice essentially depends on the diffusion coefficient and the mean grain size. Observations of the decay rate of total chlorine versus time and depth and the small grain radius (~0.1 mm) in the surface snowpack leads our estimation toward a low diffusion coefficient (~$10^{-17}$ m$^2$.s$^{-1}$). Consequently, the diffusion process may become negligible to contribute to the export of HCl from ice if the mean grain size becomes too large. We have already

noticed that the mean grain radius, estimated by the SSA, increases progressively with depth, following the general compaction process. Any other process, like the formation of depth hoar or grain metamorphism, that would re-arrange the ice mass and increase the mean grain size, have the potential to create conditions where the chlorine content of ice can virtually no longer escape the lattice. Depending on whether such microstructure rearrangement affect a snow layer at an early stage or not, it induces a large variability in the chlorine record from layer to layer. Such a variability is clearly shown by overlapping the 4

snow records at Dome C on the same depth scale (Traversi et al., 2009) (reported on Figures 3 to 7).

The two important variables of this model, the effective diffusion coefficient, and the grain size, could be related to the microstructural characteristics of the snowpack layers. The mean radius of the snow grains can be obtained by standardized techniques (Gallet et al., 2009). The effective diffusion coefficient and its variability may be more difficult to obtain. It can be

seen as a pure chemical parameter of HCl in ice, but it is potentially and partly related to the micro-structure, since post-depositional processes such as compaction and metamorphism not only affect the mean grain size but also the chlorine mobility pathways (crystal arrangement, veins…).




The four configurations obtained in Figures 7 and 8 illustrate the impact of a time lag between the formation of the snow grain
and its deposition at the core site. However, they also share the assumption of a homogeneous distribution of chlorine content
in the snow grains at the time of their formation. Any departure from this initial homogeneous distribution would lead to an
acceleration or moderation of the chlorine release from the snow grain. Determining the initial distribution of chlorine content
in snow grains is likely to be a challenging experimental effort, but relevant for determining on how fast the chlorine can be
released.


The complexity of the ice-air exchanges remains unexplained and state of the art about the possible reservoirs (grain
boundaries, veins, bulk crystals, …) and processes (pure adsorbate, temperature dependence, …) is fully commented by
(Huthwelker et al., 2006). This issue is central to the interpretation of chlorine content since the rate of release out of the ice
grains is a balance between the diffusion coefficient and the grain size, but this is only true because we have assumed a peculiar
boundary condition, which is an immediate equilibrium at the ice-air interface. This assumption is supported by the fact that
the transfer coefficients of HCl uptake at the ice-air interface are rapid relative to the observed release time scale (a few
decades). If for some reason this mass transfer coefficient at the ice-air interface were a limiting factor relative to the bulk
diffusion coefficient, the microstructural aspect of grain size would also become less relevant in explaining the temporal
dynamics. The variability of chlorine concentration observed in different snow pits (in time and space) and from layer to layer
within the same core would be explained primarily by the variability in initial snow concentration. It remains a possible option
when considering temporal variability (annual, seasonal…) but less convincing in terms of spatial heterogeneity. The latter is
best explained by microstructural variations (mean grain size, connectivity…).

The High Antarctic Plateau is subject to strong seasonal variations, expressed in variations in temperature, precipitation,
atmospheric composition, and particle inputs. Seasonal temperature cycles propagate in the first meters of the snowpack
inducing temperature gradients (Whillans and Grootes, 1985). This will affect conditions of snow deposition as well as the
conditions of metamorphism, which may be far from the annual mean average. In addition, diffusion coefficients are usually
reported to be temperature dependent (Dominé et al., 1994; Huthwelker et al., 2006). It is obvious that considering the seasonal
cycle in all these processes would be an additional step to improve the estimates made previously and should be tested in a
future work. The present work has been limited to a mean-value approach, as the objective is to propose a first-order
mechanism for the chlorine transfer function within the snowpack on a multi-decadal time scale.

Previous modelling experiments have considered detailed processes of ventilation and diffusion into the SIA, e.g. vapor
diffusion in the model CROCUS (Albert, 1996; Colbeck, 1997; Calonne et al., 2014; Touzeau et al., 2018). Ventilation is
probably not instantaneous and may occur in relation with wind intensity and change according to the seasons. During some
periods of the year, the air content of the SIA may not be similar to the above atmospheric composition, but rather reflect the



ice-air exchanges. Our little knowledge of the real composition of the snowpack interstitial air, nor of the real conditions of chemical equilibrium, do not allow us to know if this disconnection between the interstitial air and the atmospheric composition would increase or decrease the concentration of HCl gas in the SIA. In both cases, the conclusions of our study would not be
changed insofar as we made the strong assumption that in fine the concentration of HCl in the ice tends towards zero. At most, if the rate of transfer between ice and air was slow enough to be partly limiting, then this would slow down the rate of chlorine decay in the ice, without changing the final value.

An important difference between our study and previous models that have incorporated solid diffusion in ice grains is the time
scale. For example, (Touzeau et al., 2018) simulates the diffusion of water isotopes in snow grains over a period of 13 years, and because of the small value of the diffusion coefficient, they conclude that at the scale of their study solid diffusion is negligible. The specificity of the chlorine content in the High Antarctic Plateau records is to decrease over several decades, and this can be explained by a very low diffusion coefficient, compatible with theoretical studies.

**Conclusions**

At sites like Vostok and Dome C, on the High Antarctic Plateau, the chlorine content (as HCl and NaCl) in the upper meters of the snowpack shows that it is affected by post-depositional processes and is therefore in transitional state. Understanding these processes is also understanding the final record in deeper ice, which is used for many applications in paleo-climate studies.

The chlorine content of the snow that settles at these locations is composed mainly of HCl, complemented by the sea-salt component (with a $Cl^-$:$Na^+$ ratio around 0.7). We suggest that the HCl content is then diffusing in ice grains, from the interior to the ice-air interface, and is expelled to the SIA before being re-emitted in the atmosphere by the wind ventilation. We formalized these mechanisms in a model that considers snowpack layers made of similar spherical snow grains, for which mean grain size can increase with depth.

We conclude to a low value for the diffusion coefficient of HCl in ice of ~$10^{-17}$ $m^2.s^{-1}$. Such a diffusion coefficient applied to snow grains of ~0.1 mm radius at the surface of the snowpack and increasing with depth, results in timings for the release of HCl from ice to the atmosphere that are compatible with the observed decrease in total chlorine content of the snow layers.

Variability of the snow content around the mean trend can be explained by that of the snow grain size history. The microstructure of the snow layers (SSA, grain shape) and its evolution with time under different post-deposition processes (compaction, metamorphism, …) is a major aspect for driving the conditions of the chemical content preservation in the snowpack. The larger the grain size, the longer in time the chlorine content can stay in ice. Additionally, fluctuations of the





initial content of the fresh snow may be part of the signal variability. The diffusion process that is invoked takes place inside snow grains, and because of its very low diffusion coefficient combined to the increasing mean grain size of the snowpack microstructure, there is no significant mobility of chlorine from layer to layer. Coupling the bulk diffusion of chlorine in snow grains with more realistic models of evolution of the snow microphysics would be a step toward coupled snowpack models. Validation of the link between chemistry and microstructure needs systematic measurements of microstructural characteristics of the snow layers in addition to chemical measurements, especially the SSA.

The HCl content of the snow is progressively released from the snowpack to the atmosphere and is made available for being captured again during the fresh snow formation. Chlorine is somehow recycled at large scale, which increases its concentration both in the Antarctic atmosphere and in snow, and its residence time over the High Antarctic Plateau. This mechanism can be applied to the $^{36}$Cl nuclide. The reconstruction of the historical anthropogenic $^{36}$Cl content in snow deposit, that we have presented, is based on the assumption that this nuclide arrives as H$^{36}$Cl form. Consequently, its fate is to be released from the snowpack to the atmosphere, and be re-incorporated into snow, like HCl. The surface composition at Vostok in 2008 shows that the fresh snow is contaminated by anthropogenic $^{36}$Cl and that its concentration is decreasing with time due to a progressive dilution of the anthropogenic signal over the decades. Measurements of the $^{36}$Cl composition of the nowadays surface elements (fresh snow, gaseous and aerosol phases) would be a precious information to constrain the time during which the recycling processes operate on the High Antarctic Plateau.

## Appendices

### Appendix A. Evaporation condition at the surface of the sphere

Equations (6.39) to (6.43) from (Crank, 1979) to get the concentration in the sphere in the case of evaporation conditions at the ice-air interface. We recall equation (6.39) about the evaporation condition at the surface of the sphere:

$$-D \ \partial C / \partial r = \ \alpha \ (C_s - \ C_\infty) \tag{S1}$$

where D is the diffusion coefficient, $\alpha$ is the mass transfer coefficient, $C_\infty$ the equilibrium concentration, $C_s$ the actual concentration just within the sphere.

### Appendix B. Optimization process

The optimization procedure consists in fixing a certain value for the diffusion coefficient D (from $0.2 \ 10^{-17}$ to $1.5 \ 10^{-17}$ m$^2$.s$^{-1}$, with an increment of $0.5 \ 10^{-18}$ m$^2$.s$^{-1}$) as well as the time lag offset tlag (taking the values of 0, 4, 8 and 12 months, successively). For each of these combinations, values of $C_0$ are tested, producing a profile of concentration versus depth. The difference



between this result and the dataset from (Traversi et al., 2009) is quantified by a RMSD (root mean square of differences). A gradient optimization process (solver Nelder-Mead from scipy.optimize.minimize package (Gao and Han, 2012)) allows to find the value of $C_0$ that minimizes this RMSD. Figure S1 shows the result of the optimization process.

The optimum coefficient diffusion is found between 1 and 1.2 $10^{-17}$ m$^2$.s$^{-1}$, depending on the time lag considered. This narrow range of values makes the estimation more robust. The RMSD value also increases rapidly as the diffusion coefficient moves away from this central value.

**Appendix C. Formula for reconstructing the H$^{36}$Cl concentration at deposition**

Considering that H$^{35}$Cl and H$^{36}$Cl undergo the same processes in the snowpack, their ratio in snow layers (at time $t$) is equal

to the one at the time of deposition (dep):

$$\frac{H^{36}Cl_{dep}}{H^{35}Cl_{dep}} = \frac{H^{36}Cl_t}{H^{35}Cl_t} \tag{S2}$$

And

$$H^{36}Cl_{dep} = H^{36}Cl_t \frac{H^{35}Cl_{dep}}{H^{35}Cl_t} \tag{S3}$$

If we assume that all $^{36}$Cl is in the HCl form, i.e., $Na^{36}Cl_t = Na^{36}Cl_{dep} = 0$, then $^{36}Cl_t = H^{36}Cl_t$.

The H$^{35}$Cl content of the snow layer can be estimated based on the Cl$^-$:Na$^+$ ratio in the aerosol deposit, so that

$$H^{35}Cl_t = {}^{35}Cl_t - Na^{35}Cl_t * \left(\frac{Cl}{Na}\right)_t \tag{S4}$$

Finally, the $^{36}$Cl content of the snow deposit can be expressed as follow:

$$^{36}Cl_{dep} = H^{36}Cl_{dep} = {}^{36}Cl_t \frac{H^{35}Cl_{dep}}{{}^{35}Cl_t - Na^{35}Cl_t * \left(\frac{Cl}{Na}\right)_t} \tag{S5}$$

**Code availability**

The model code is made available on Zenodo (Giraud, 2022) or upon request to the corresponding author.

**Author contribution**

X. Giraud: conceptualization, methodology and model creation, formal analysis, writing original draft, review and editing. M. Baroni: Funding acquisition, conceptualization, data resources, writing, review and editing. R. Traversi: data resources, review and editing.



**Competing interests**

The authors declare that they have no conflict of interest.

**Acknowledgements**

This work has been supported by the French CNRS - LEFE - PROXYNNOV project, the European Union's Horizon 2020 research and innovation program, under grant agreement no. 815384 through the project Beyond EPICA-Oldest Ice. We thank
the logistical support from the IPEV and the AARI.

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





**Table 1: Simulations summary**

| Simulation name | D (m².s⁻¹) | C₀ (ng.g⁻¹) | tlag (months) | C_dep (ng.g⁻¹) | r₀ (mm) | grain size evolution | Illustrated in Figures |
|---|---|---|---|---|---|---|---|
| standard | $10^{-17}$ | 215 | 0 | 215 | 0.1 | standard (b = 0.5) | 4, 5, 6, 7, 8 |
| constant_radius_low | " | " | " | " | 0.1 | constant | 4 |
| constant_radius_high | " | " | " | " | 0.4 | constant | 4 |
| minor_radius | " | " | " | " | 0.1 | minor increase (b = 0.3) | 4 |
| major_radius | " | " | " | " | 0.1 | major increase (b = 0.65) | 4 |
| large_radius | " | " | " | " | 0.4 | standard | 4 |
| low_Diff | $0.5\ 10^{-17}$ | 215 | 0 | 215 | 0.1 | standard | 5 |
| high_Diff | $2\ 10^{-17}$ | " | " | " | 0.1 | standard | 5 |
| low_Conc0 | $10^{-17}$ | 100 | 0 | 100 | 0.1 | standard | 6 |
| high_Conc0 | " | 300 | " | 300 | 0.1 | standard | 6 |
| time_lag_4_months | $1.1\ 10^{-17}$ | 290 | 4 | 195 | 0.1 | standard | 7, 8 |
| time_lag_8_months | $1.15\ 10^{-17}$ | 345 | 8 | 190 | 0.1 | standard | 7, 8 |
| time_lag_12_months | $1.2\ 10^{-17}$ | 410 | 12 | 190 | 0.1 | standard | 7, 8 |
| Vostok_standard | $10^{-17}$ | 440 | 0 | 440 | 0.1 | standard | 9 |
| Vostok_varying_radius | $10^{-17}$ | " | 0 | " | 0.1 | adjusted (see section 4) | 9 |
| Vostok_varying_C0 | $10^{-17}$ | adjusted | 0 | adjusted | 0.1 | standard | 9 |



**List of Figures**

**Figure 1:** HCl concentration profiles on the High Antarctic Plateau. a) Vostok from (from (Pivot et al., 2019)), versus depth; b) Dome C (from (Traversi et al., 2009)), versus depth. c) Same records versus time, assuming an accumulation rate of 8 cm.yr$^{-1}$ at Dome C.

**Figure 2:** Snow grain radius versus depth as reported at Dome C by (Gallet et al., 2011) (black dotted line), and from an 710 idealised formula used in the model (blue line).

**Figure 3:** Theoretical continuity of diffusion out of a sphere as computed in the model in case of changing spherical grain radius. a) Grain radius. b) Total chlorine concentration.

**Figure 4:** Effect of the evolution of the grain size. Constant radius at 0.1 mm (simulation: constant_radius_low; brown lines); constant radius at 0.4 mm (simulation constant_radius_high; turquoise lines); standard increase of the grain radius (simulation: standard; green lines); moderate increase of the grain radius (simulation: minor_radius; yellow lines); major increase of the grain radius (simulation: major_radius; red lines); grain radius start at 0.4 mm (simulation large_radius; black lines); The thin black line in panel (b) is the exact analytical solution provided by the equations of (Crank, 1979). See Table 1 for the parameters 720 of each simulation.

**Figure 5:** Effect of the diffusion coefficient. D = 5.10$^{-18}$ m$^2$.s$^{-1}$: light green (simulation low_Diff); D = 10$^{-17}$ m$^2$.s$^{-1}$: green (simulation standard); D = 2.10$^{-17}$ m$^2$.s$^{-1}$: dark green (simulation high_Diff).

**Figure 6:** Effect of the initial concentration of chlorine in snow grains. C$_0$ = 100 ng.g$^{-1}$: yellow (simulation low_Conc0); C$_0$ = 215 ng.g$^{-1}$: green (simulation standard); C$_0$ = 300 ng.g$^{-1}$: olive (simulation high_Conc0).

**Figure 7:** Effect of the time lag between formation and deposition of snow grains.

**Figure 8:** Chlorine distribution into the spherical snow grains at deposition time.

**Figure 9:** Modelling of chlorine content at Vostok and its variability. a) Initial concentration of chlorine in snow grains. b) Evolution of the grain radius. c) Snow content of total chlorine (thin blue line) and total sodium (thin yellow line) as presented by (Pivot et al., 2019). The simulation 'Vostok_standard' (thick green lines) consider a constant initial concentration of snow 735 grains and a monotonic increase of the grain size in each layer. the The simulation 'Vostok_varying_C0' (thick orange linse) considers a monotonic increase of the grain radius with depth, and the variability lies in the initial concentration at deposition.





The simulation 'Vostok_varying_radius' (thick purple lines) considers constant initial concentrations of the snow, and different evolutions of the snow grain sizes for each layer. Both simulations 'Vostok_varying_C0' and 'Vostok_varying_radius' produce profiles of HCl versus depth that fits to the estimation of the data HCl content in the snowpit.


**Figure 10:** History of the total $^{36}$Cl content in the snow deposit as reconstructed for the record from (Pivot et al., 2019) at Vostok with the simulation 'Vostok_varying_radius': panels a) and b) show the same data on different y scales. Chlorine isotope data as presented in (Pivot et al., 2019) and used for this simulation : c) $^{36}$Cl:Cl ratio. d) $^{36}$Cl concentration.



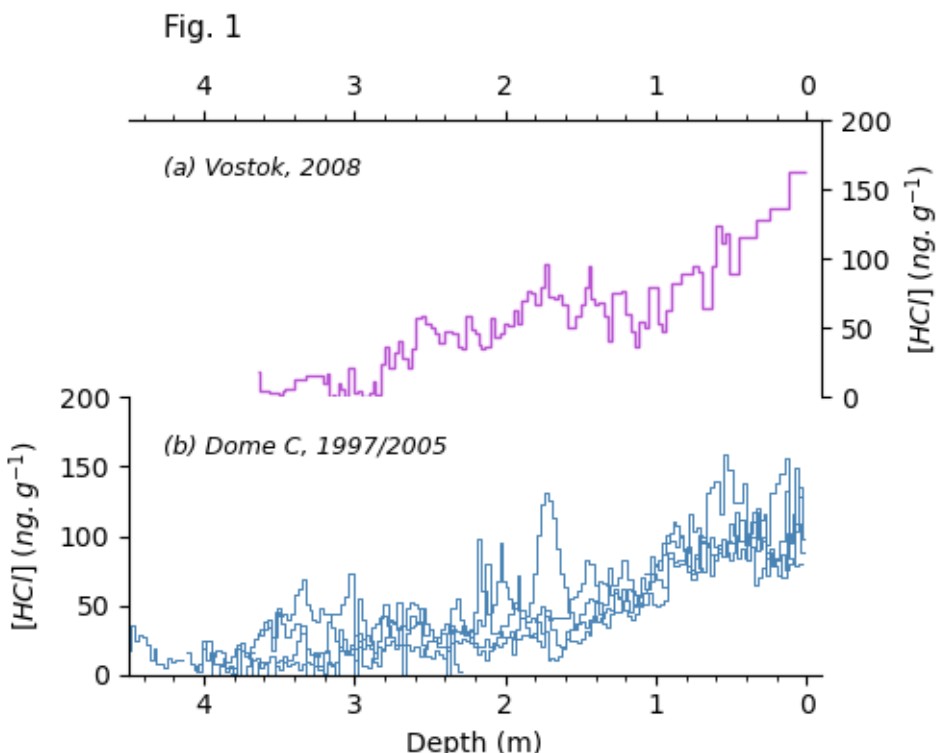

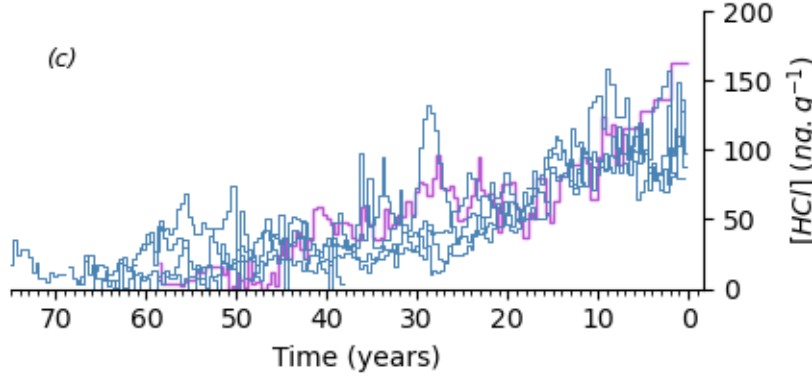






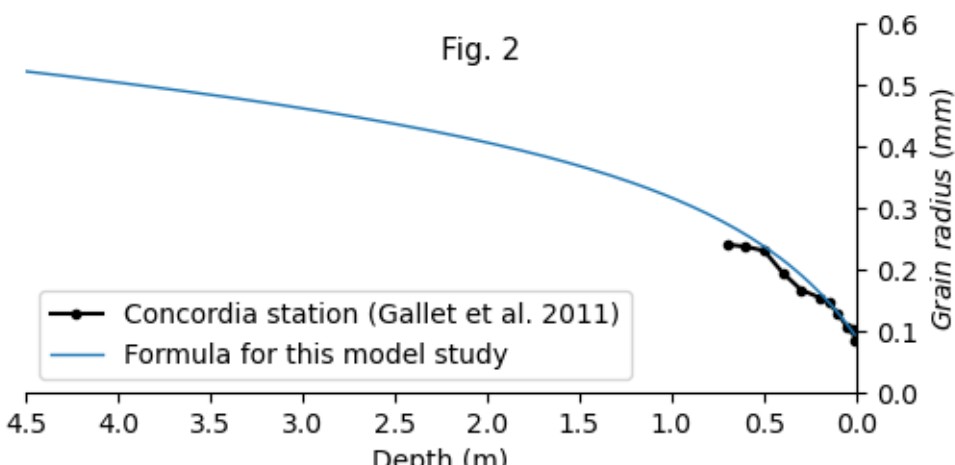



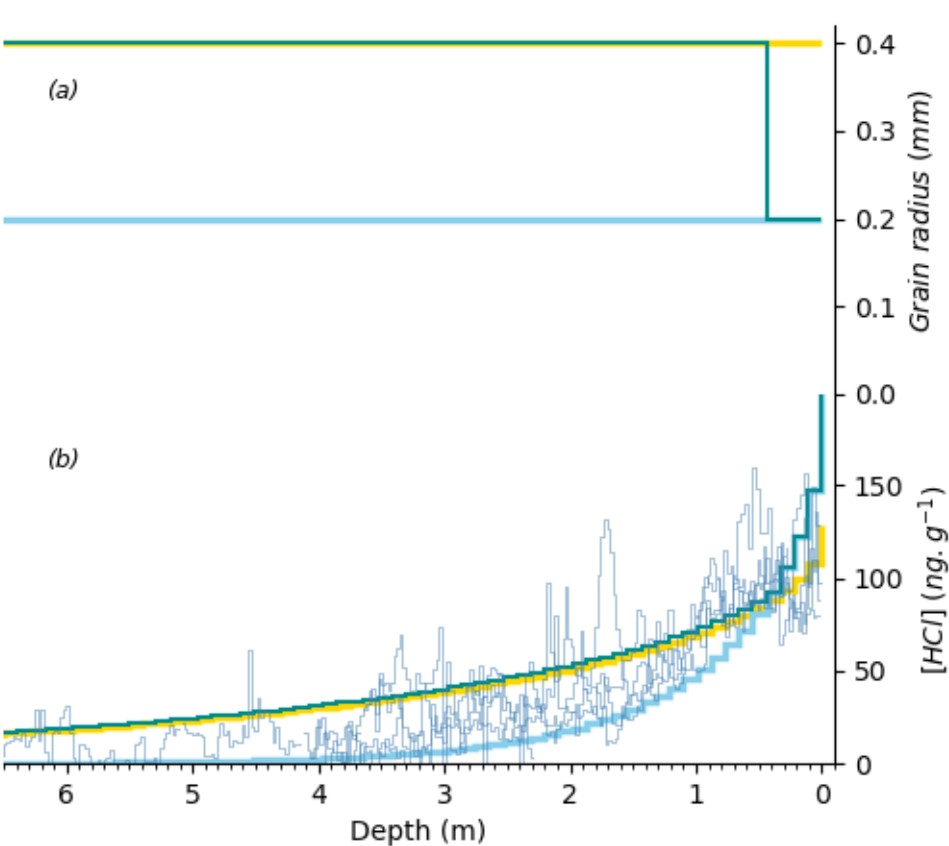

Fig. 3




Fig. 4

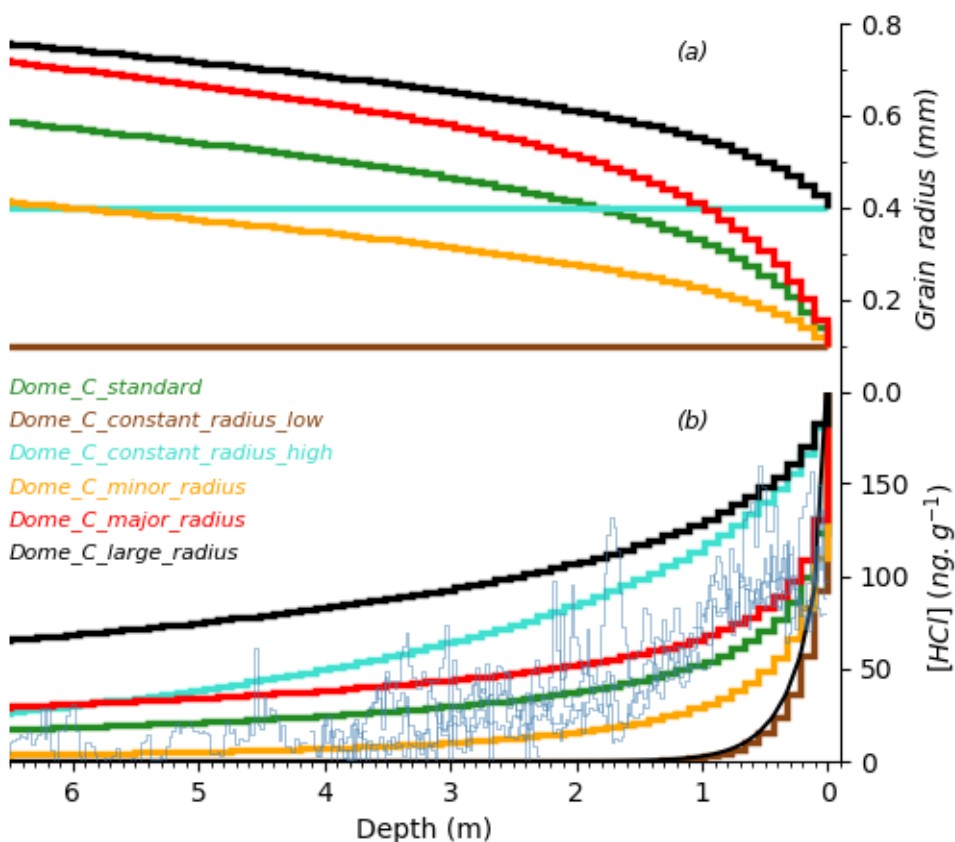



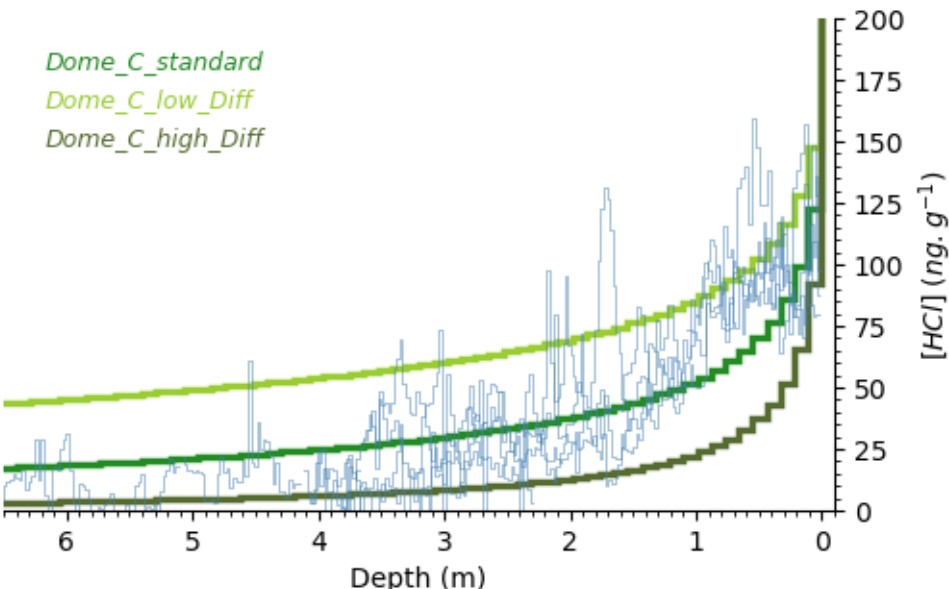



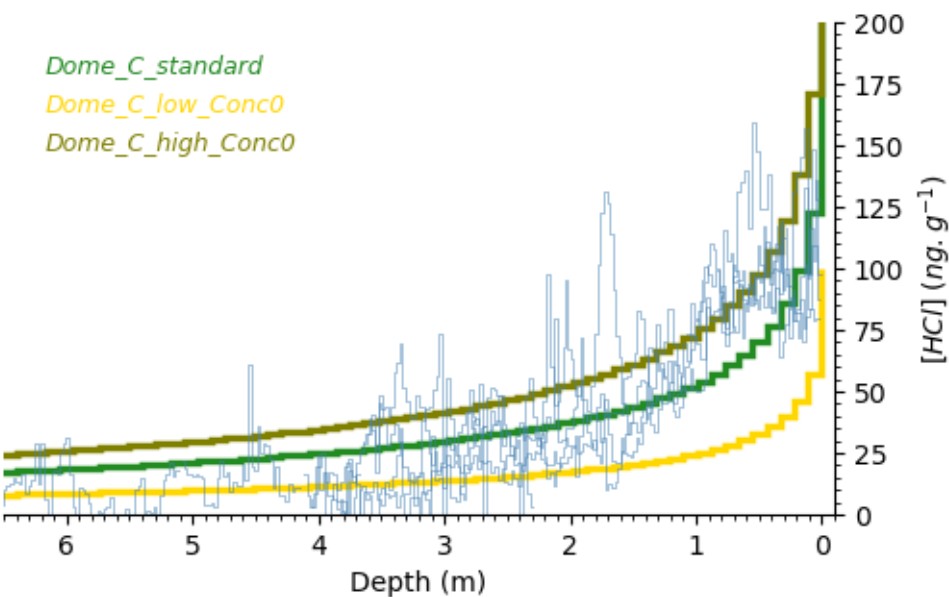

Fig. 6






Fig. 7

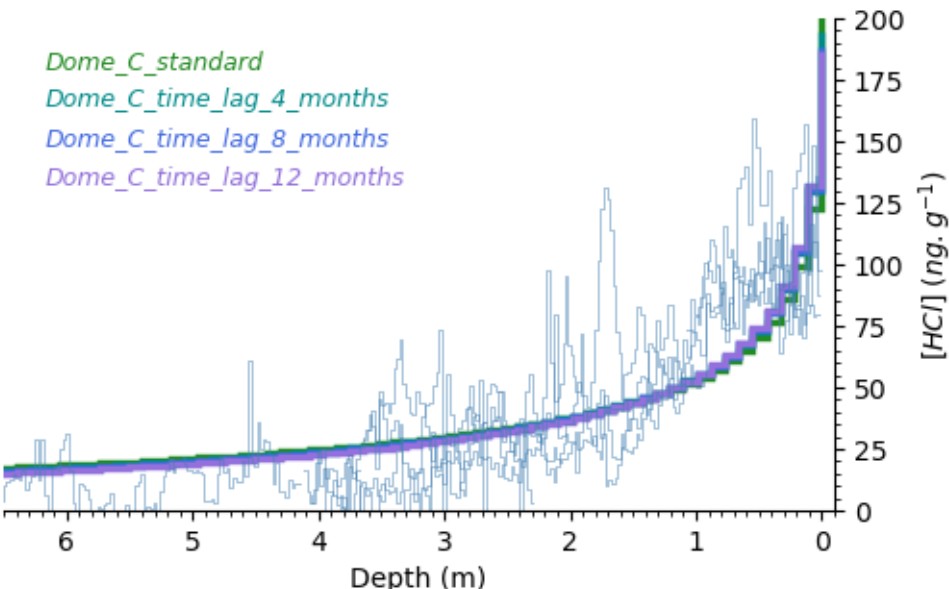





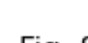

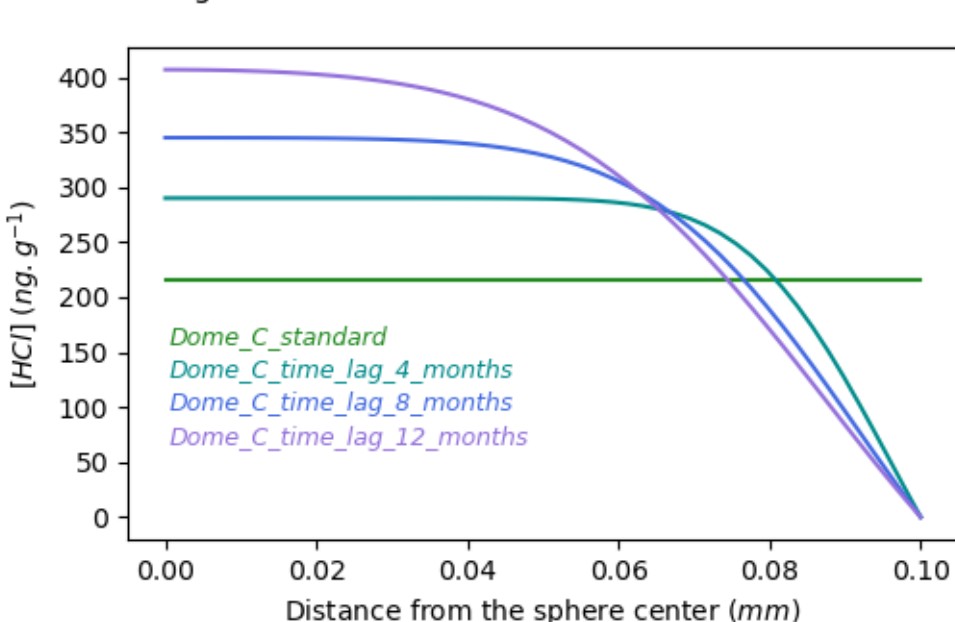




Fig. 9





Fig. 10