# Peer review of "A mechanism of post-depositional processes affecting chlorine and its isotope in the upper snowpack of High Antarctic Plateau"

_EGUsphere, 2022_

## Referee Comment (RC1)

**Review of "A mechanism of post-depositional processes affecting chlorine and its isotope in the upper snowpack of High Antarctic Plateau" By Giraud et al.**

Reviewer: Florent Domine

The authors propose a model to explain the observed decrease in chloride concentration with depth in firn cores of the East Antarctic ice sheet, in regions with low snow accumulation (Dome C and Vostok). Based on the Cl/Na ratio in these cores, they infer that a small fraction of the chloride is in the NaCl form, i.e. in a particulate phase, while most of the chloride is present as HCl as a solute in ice crystals. The fundamental hypothesis of their model is that HCl is mobile and diffuses out of the ice crystals into the snowpack interstitial air (SIA) where it is transferred to the atmosphere by windpumping. This explains the loss of HCl and therefore the decrease of the chloride signal to very low values, with Cl/Na=0.7, typical of the composition of the sea salt aerosol on the plateau. HCl is therefore the mobile phase that is totally released from the firn, while NaCl is immobile and retained in the firn.

Another important hypothesis is "we will consider the SIA content to be homogeneous over that depth interval [top 3-5 m of the snowpack]". From this, given that there does not seem to be any HCl at 5 m depth, the authors deduce that "it implies that the ice-air equilibrium value for the mobile and volatile form of chlorine (HCl) at the surface of the snow grain is close to zero."

This seems quite problematic. First of all, the authors never consider temperature as a variable, while almost all thermodynamic and kinetic processes are temperature-dependent. There are significant yearly temperature fluctuations at Dome C and Vostok, and there is a significant temperature gradient in the firn, which probably should be considered. But the most important aspect is that the ice-air equilibrium is most likely not close to zero. (Thibert and Domine, 1997) have made a detailed investigation of this equilibrium, as a function of the partial pressure of HCl and of temperature. This equilibrium study seems to have stood the test of time and as far as I know has never been seriously questioned. The thermodynamic parameters of this equilibrium, such as the partial molar enthalpy of mixing of HCl in ice are sensible and in fact close to what basic thermodynamic considerations should expect. There is therefore no valid reason to ignore this work (even though the paper is cited, but only for the diffusion coefficient measured) and to state that the equilibrium should be close to zero, simply based on questionable and little-justified hypotheses.

(Legrand et al., 2017) (cited by the authors) in their Figure 3 show that HCl concentrations in the air near Dome C are about 50 ng m$^{-3}$ in summer and 5 ng m$^{-3}$ in winter. Assuming T=200K in winter and T=240K in summer, the equilibrium expression of (Thibert and Domine, 1997) indicates that in surface snow the chloride concentration (excluding the contribution from particulates, which are another phase and therefore are not considered in the equilibrium) should be 5540 ng g$^{-1}$ in winter and 1328 ng g$^{-1}$ in summer. Surface snow is not necessarily in equilibrium with the atmosphere, as suggested e.g. at Summit, where it was found to be undersaturated (Domine et al., 1995), but these values nevertheless illustrate the impact of equilibrium values on concentrations. This equilibrium also implies that, if there is no chloride as HCl in the firn at depth, the HCl concentration in the SIA is near zero. This is contrary to an important hypothesis of the model proposed, as noted above: "we will consider the SIA content to be homogeneous over that depth interval [top 3-5 m of the snowpack]". In any case, given that there is a vertical chloride concentration gradient in the snow, suggesting that there is no concentration gradient in the SIA is rather untenable. Furthermore, suggesting that there is loss of HCl from the firn to the surface

air implies a flux out of the firn, and therefore, as dictated by basic physics, there must be a vertical concentration gradient in the SIA.

The fundamental hypotheses of the model (equilibrium near zero and homogeneous SIA) therefore do not seem reasonable. Furthermore, the system used by the author is underdetermined, meaning that, even if the model were correct, testing it would be very difficult, as the authors acknowledge (line 341) "Figure 7 illustrates that it is possible to find different combinations of initial concentration, $C_0$, and time lag before deposition, $t_{lag}$, to achieve similar concentrations at deposition time, $C_{dep}$, and similar evolutions of chlorine content with depth."

Given all these critical flaws, I must come to the conclusion that I cannot recommend publication of this paper in The Cryosphere, even after major revisions.

The observations that the concentration of mobile chloride in firn cores decreases with depth is nevertheless important and deserves attention. It is interesting that, based on the equilibrium of (Thibert and Domine, 1997), snow at Dome C and Vostok appears undersaturated with respect to HCl in the atmosphere. The equilibrium then dictates that HCl should diffuse into ice crystals, not out of them, and that, with HCl input into the firn air by windpumping, chloride in snow should increase with depth.

At present, I can only see 2 possibilities to explain this quandary. The first one is that the equilibrium of (Thibert and Domine, 1997) is totally wrong, which I personally do not believe. Note that even if the equilibrium was wrong, this would not be sufficient to make the model valid, as other hypotheses pose problems, as mentioned above. The second one is that there is a loss process of HCl in the firn, and that this loss process is not detected by the analytical protocol used by (Traversi et al., 2009) and many others, who used ion chromatography on melted and filtered snow samples.

I therefore wish to tentatively offer an alternate hypothesis to explain the decrease in the chloride signal in firn cores measured by IC. Based on thermodynamics, this loss process has to involve another phase and the only reasonable possibility is probably mineral dust. (Baccolo et al., 2021) have made a very interesting observation in ice from Talos Dome. They have convincingly demonstrated that there are some solid phase reactions within the mineral dust particles, which induce mineralogical changes with depth. They propose that "The expulsion of acidic atmospheric species from ice grains and their concentration in localized environments is likely the main process responsible for englacial reactions." I therefore propose that some mineral dust particles in firn at Dome C and Vostok react with HCl present in the SIA. The SIA therefore becomes depleted in HCl and this induces the diffusion of HCl out of the ice grains. Eventually, ice grains become almost completely depleted in HCl, as most or all of this HCl has reacted with particles. The HCl depletion is therefore not because the air-ice equilibrium is nearly zero, but because HCl is consumed by reactions with minerals. If these minerals are not soluble, the chloride in these minerals will not be detected by IC, and it will seem like the total chloride signal decreases with depth. In fact, the signal is probably fairly constant, but most of the chloride is trapped in a phase where it is not analyzed. Testing this hypothesis requires measuring chloride in firn samples by a method other than IC. Atomic absorption without filtration might work. Other methods, including some based on mass spectrometry, may also work. Testing this, however, does not appear possible with available measurements.

**References**

Baccolo, G., Delmonte, B., Di Stefano, E., Cibin, G., Crotti, I., Frezzotti, M., Hampai, D., Iizuka, Y., Marcelli, A., and Maggi, V.: Deep ice as a geochemical reactor: insights from iron speciation and mineralogy of dust in the Talos Dome ice core (East Antarctica), The Cryosphere, 15, 4807-4822, 2021.

Domine, F., Thibert, E., Silvente, E., Legrand, M., and Jaffrezo, J. L.: Determining past atmospheric HCl mixing ratios from ice core analyses, J. Atmos. Chem., 21, 165-186, 1995.

Legrand, M., Preunkert, S., Wolff, E., Weller, R., Jourdain, B., and Wagenbach, D.: Year-round records of bulk and size-segregated aerosol composition in central Antarctica (Concordia site) Part 1: Fractionation of sea-salt particles, Atmos. Chem. Phys., 17, 14039-14054, 2017.

Thibert, E. and Domine, F.: Thermodynamics and kinetics of the solid solution of HCl in ice, J. Phys. Chem. B, 101, 3554-3565, 1997.

Traversi, R., Becagli, S., Castellano, E., Cerri, O., Morganti, A., Severi, M., and Udisti, R.: Study of Dome C site (East Antartica) variability by comparing chemical stratigraphies, Microchem. J., 92, 7-14, 2009.